# Phenylalanine Increases the Production of Antioxidant Phenolic Acids in *Ginkgo biloba* Cell Cultures

**DOI:** 10.3390/molecules26164965

**Published:** 2021-08-17

**Authors:** Agnieszka Szewczyk, Inga Kwiecień, Mariusz Grabowski, Karolina Rajek, Emilia Cavò, Maria Fernanda Taviano, Natalizia Miceli

**Affiliations:** 1Department of Pharmaceutical Botany, Faculty of Pharmacy, Jagiellonian University Medical College, 30-688 Krakow, Poland; inga.kwiecien@uj.edu.pl; 2SSG of Medicinal Plants and Mushroom Biotechnology Department of Pharmaceutical Botany, Jagiellonian University Medical College, Medyczna 9 Str., 30-688 Cracow, Poland; mgrabowski5@op.pl (M.G.); karolina.rajek@student.uj.edu.pl (K.R.); 3Department of Chemical, Biological, Pharmaceutical and Environmental Sciences, University of Messina, Viale Palatucci, 98168 Messina, Italy; ecavo@unime.it (E.C.); mtaviano@unime.it (M.F.T.); nmiceli@unime.it (N.M.)

**Keywords:** *Ginkgo biloba*, in vitro cultures, phenolic acids, antioxidant activity, *Artemia salina*, phenylalanine

## Abstract

The aims of this study were to evaluate the antioxidant properties, to investigate the content of major secondary metabolites in *Ginkgo biloba* cell cultures, and to determine the change in the production of phenolic acids by adding phenylalanine to the culture medium. Three in vitro methods, which depend on different mechanisms, were used for assessing the antioxidant activity of the extract: 1,1-diphenyl-2-picrylhydrazil (DPPH), reducing power and Fe^2+^ chelating activity assays. The extract showed moderate activity both in the DPPH and in the reducing power assays (IC_50_ = 1.966 ± 0.058 mg/mL; ASE/mL = 16.31 ± 1.20); instead, it was found to possess good chelating properties reaching approximately 70% activity at the highest tested dose. The total phenolic, total flavonoid, and condensed tannin content of *G. biloba* cell culture extract was spectrophotometrically determined. The phenolic acid content was investigated by RP-HPLC, and the major metabolites—protocatechuic and *p*-hydroxybenzoic acids—were isolated and investigated by ^1^H NMR. The results showed that phenylalanine added to *G. biloba* cell cultures at concentrations of 100, 150, and 200 mg/150 mL increased the production of phenolic acids. Cultures that were grown for 3 weeks and collected after 4 days of phenylalanine supplementation at high concentration showed maximal content of phenolic acids (73.76 mg/100 g DW).

## 1. Introduction

*Ginkgo biloba* L. (Ginkgoaceae) is the only survivor of the order Ginkgoales and is considered a “living fossil”. The leaves of this plant contain di- and sesquiterpene lactones and flavonoids that are biologically active. Phytopharmaceuticals derived from *G. biloba* leaves are used in medicine, especially in the treatment of cerebral and peripheral disorders. The leaves extract possesses antioxidative, radical scavenging and neuroprotective properties. Active compounds in *G. biloba* extract improve blood circulation, prevent clot formation, reinforce the walls of the capillaries, and protect nerve cells from injury when they are deprived of oxygen [1,2]. Various studies have investigated *G. biloba* for the presence of different classes of chemical constituents, and the results revealed that this plant contains a number of secondary metabolites, including terpenoids, polyphenols, allyl phenols, organic acids, carbohydrates, fatty acids and lipids, amino acids, and inorganic salts [3,4,5,6]. Plant cell biotechnology studies on *G. biloba* focused on micropropagation or biosynthesis and accumulation of ginkgolides and bilobalide [7,8,9,10,11,12,13,14]. The other secondary metabolites, such as catechins, were rarely investigated in in vitro cultures [15]. The content of phenolic acids in *G. biloba* in vitro cultures has not been investigated yet. Phenolic acids exhibit antioxidant activity due to a multidirectional mechanism of action. They are characterized by the ability to inactivate free radicals, break radical chain reactions, inhibit oxidase activity, chelate metal ions included in the enzyme catalyzing oxidation reactions, and reduce properties. Phenolic acids also show anti-inflammatory, antimicrobial, choleretic, and immunostimulatory effects [16]. The plant *G. biloba* is widely used in medicine and has also gained great attention in the cosmetic industry. The enormous demand for this plant material has sparked interest in various plant biotechnology methods as a means of obtaining secondary metabolites from the species. Undifferentiated plant in vitro systems have recently been increasingly used in the cosmetics industry. Most of the cosmetic ingredients obtained by in vitro technology are based on the cultivation of callus or cell cultures. Cell cultures make it possible to scale up the culture in various types of bioreactors. Examples of undifferentiated in vitro cultures currently used in the cosmetics industry are *Symphytum officinale*, *Saponaria pumila*, *Morinda citrifolia,* and *Gossypium herbaceum* callus cultures. Water extracts from cultures are used in skin care products [17]. In vitro plant culture extracts contain a mixture of bioactive substances belonging to primary and secondary metabolites. The cosmetics industry often uses extracts from in vitro cultures, containing ingredients with antioxidant properties, such as phenolic compounds (*Calendula officinalis*, *Silybum marianum*, *Centella asiatica,* and *Rubus chamaemorus*). It is worth mentioning that extracts from in vitro cultures are used in cosmetics in low concentrations [18]. Plants grown in vitro often produce a lower amount of secondary metabolites than plants grown in soil; however, biotechnology offers a number of strategies for increasing the production of active compounds (e.g., selection of appropriate plant growth and development regulators, change in medium composition, change in the physical conditions of the culture, elicitation, and addition of metabolic pathway precursors) [19].

In this study, callus cultures were initiated from the leaves of female *Ginkgo biloba* specimens and cell cultures were derived from the callus cultures. The antioxidant properties of the extract obtained from *G. biloba* cell cultures were determined by different assays, based on different mechanisms, as well as the assessment of the toxicity using the brine shrimp (*Artemia salina* Leach) lethality bioassay. Total phenolic, flavonoid, and condensed tannin content were determined, too. In the next step, *Ginkgo biloba* cell cultures were investigated to test their ability to accumulate phenolic acids and to increase the accumulation of selective metabolites by adding phenylalanine, a metabolic pathway precursor.

## 2. Results and Discussion

### 2.1. Antioxidant Activity

The antioxidant effect is, inter alia, associated with an increased level of superoxide dismutase, catalase, glutathione, and enzymatic activity of cytochrome P-450. Stefanovits-Bảnyai et al. (2006) investigated the antioxidant activity of the leaf extracts of female and male *G. biloba* specimens. The results obtained revealed that the male leaf extract showed greater free radical-scavenging activity, whereas the female leaf extract contained more ions (Mg^2+^, Ca^2+^, K^+^, Na^+^, and Zn^2+^) and thus showed a positive effect on degenerative brain diseases. At a concentration of 10–500 µg/mL, the *G. biloba* extract showed protection against free radicals damage to DNA structures and cell membranes of human lymphocytes. The antioxidant activity exhibited by the leaf extracts was comparable to that of α-tocopherol [20].

Antioxidants can be classified either as primary or chain-breaking, which actively inhibit oxidation reactions, and secondary or preventive, which inhibit oxidation indirectly.

According to their antioxidant mechanism, much of primary antioxidant chemistry reactions can be grouped into the categories of hydrogen-atom transfer (HAT) and single-electron transfer (SET). The HAT mechanism occurs when an antioxidant compound scavenges free radicals by donating hydrogen atoms; on the other hand, when an antioxidant transfers a single electron to reduce any compound, including metals, carbonyls, and free radicals, the SET mechanism occurs [21,22]. Several authors state that although many antioxidant reactions are characterized by following HAT or SET chemical processes, these reaction mechanisms can occur simultaneously [22,23,24].

Secondary antioxidants act through several mechanisms, including the chelation of metal ions capable of catalyzing oxidative processes and scavenging oxygen, absorbing UV radiation, inhibiting enzymes, or decomposing hydroperoxides [25]. Since antioxidant activity is mediated by different mechanisms, no single testing method can provide a comprehensive view of a sample’s antioxidant profile. Therefore, various methods based on diverse approaches and mechanisms must be used to assess the antioxidant activity of plant-derived phytocomplexes or isolated compounds. In this study, three in vitro tests based on different mechanisms were utilized to investigate the antioxidant effectiveness of *G. biloba* cell culture extract: the primary antioxidant activity was examined using the DPPH assay, based on both HAT and SET mechanisms, and the reducing power assay was based on an SET-based method; the secondary antioxidant properties were determined by the ferrous ion (Fe^2+^) chelating activity assay.

Figure 1 shows the results of the DPPH assay, utilized to determine the free scavenging ability of *G. biloba* cell culture extract. Compared with the reference standard BHT, it displayed lower activity in the range of concentrations tested, which increased with increasing dose. The extract showed about 50% activity at the highest concentration tested. As indicated by the half maximal inhibitory concentration (IC_50_) values, the extract exhibited a moderate radical scavenging activity compared to BHT (1.97 ± 0.06 mg/mL and 0.06 ± 0.01 mg/mL, respectively).

The results of reducing power assay of *G. biloba* cell culture extract, determined through the Fe^3+^–Fe^2+^ transformation method, showed that it displayed mild activity compared with the standard BHT as confirmed by ASE/mL values (16.31 ± 1.20 ASE/mL and 0.89 ± 0.06 ASE/mL, respectively) (Figure 2).

In the Fe^2+^ chelating activity assay, performed by evaluating the inhibiting effect on the Fe^2+^–ferrozine complex formation, *G. biloba* cell culture extract exhibited good chelating properties that were dose-dependent, although lower than those of the reference standard EDTA, as confirmed by the IC_50_ values (1.26 ± 0.04 mg/mL and 0.01 ± 3.55 × 10^−5^ mg/mL, respectively) (Figure 3).

The results of the antioxidant tests indicated that the *G. biloba* cell culture extract possesses both primary and secondary antioxidant properties, with the latter being stronger than the former. Polyphenol compounds are the major antioxidant plant secondary metabolites [26]. The antioxidant properties of polyphenols have been widely demonstrated; these compounds can act either as free radical scavengers, reducing agents, or metal chelators [27]. Flavonoids and phenolic acids are the largest classes of plant phenolics; several compounds from these classes have been shown to possess powerful antioxidant activity in both in vitro and in vivo investigations [25,28]. In order to establish the relationship between the observed antioxidant activity and the polyphenols contained in the *G. biloba* cell culture extract, their amount was determined.

### 2.2. Content of Secondary Metabolites

#### 2.2.1. Total Phenolic, Flavonoid, and Condensed Tannin Content

Colorimetric reactions are widely used in the UV/VIS spectrophotometric method, which is easy to perform, rapid, applicable in routine laboratory use, and low-cost. The Folin–Ciocalteu assay is widely used for determination of total phenolics; this assay relies on the transfer of electrons in alkaline medium from phenolic compounds to phosphomolybdic/phosphotungstic acid reagent to form blue colored complexes that are determined spectrophotometrically at 760 nm. Numerous examples of the application of this assay to characterize natural products may be found in the literature. In most cases, total phenolics quantified by this method are correlated with the antioxidant capacities confirming the value of the Folin-Ciocalteau test [29]. In this study, the total number of phenolics in *G. biloba* cell culture extract was found to be equal to 42.10 ± 0.11 mg GAE/g extract. In the vanillin assay, used to measure total proanthocyanidin (condensed tannin) content, condensation of resorcin- or phloroglucin-partial structure of flavonols with vanillin in acidic medium leads to the formation of colored carbonium ions. On the other hand, complexation of the phenolics with Al(III) is the principle of spectrophotometric assay used for quantification of total flavonoids [30]. Regarding the amounts of total flavonoids and condensed tannins contained in *G. biloba* cell culture extract, they were found equal to be 8.21 ± 0.48 mg QE/g extract and 4.90 ± 0.88 mg CE/g extract, respectively.

#### 2.2.2. Content of Phenolic Acids

From the spectrophotometric determinations of the polyphenolic compounds, it has been found that, of the total quantity of polyphenols produced by *G. biloba* cell culture, only a small amount is represented by the flavonoids. This could explain the moderate primary antioxidant activity highlighted for the methanol extract. Nevertheless, the cell culture extract exhibited good chelating activity; therefore, it could be hypothesized that, although flavonoids might contribute to some extent, other antioxidant phytochemicals besides flavonoids were responsible for this effect. HPLC analysis revealed the presence of phenolic acids, which are thought to be the main secondary metabolites involved in the secondary antioxidant activity of the extract. Previous investigations [31] have shown that protocatechuic, *p*-hydroxybenzoic, and vanillic acids were the main phenolic acids found in the leaves. Seven free and liberated-by-hydrolysis phenolic acids were identified as protocatechuic, *p*-hydroxybenzoic, vanillic, caffeic, *p*-coumaric, ferulic, and chlorogenic acids. Studies showed that the concentration of free phenolic acids amounted to 19.693 µg/g of fresh leaves [32,33].

This study detected the presence of protocatechuic acid, *p*-hydroxybenzoic acid, gallic acid, and trace amounts of vanillic acid in methanol extracts of leaf, callus, and cell cultures (Table 1). The main phenolic acids detected in the female leaves were protocatechuic and *p*-hydroxybenzoic acids. The same phenolic acids dominated in in vitro cultures as well.

The major phenolic acids (protocatechuic and *p*-hydroxybenzoic acids) present in in vitro cultures were isolated by column and thin-layer chromatography, and their structures were initially confirmed by ^1^H NMR (see Appendix A).

All the phenolic acids detected belong to the class of hydroxybenzoic acids, whose iron chelating ability has been previously reported. The results of a study performed by Andjelković, et al. (2006) to estimate and compare the iron chelating ability of various phenolic acids showed that, among hydroxybenzoic acids, those bearing catechol or galloyl groups, such as protocatechuic acid and gallic acid, showed complex formation, whereas those without these groups, such as vanillic acid, did not show any complex formation. Moreover, gallic acid, with galloyl moiety, was found to be a stronger chelator than protocatechuic acid bearing a catechol moiety [34]. Indeed, some investigations carried out on medicinal plant extracts have correlated the observed iron chelating activity with their relative abundance of protocatechuic, *p*-hydroxybenzoic, and gallic acids [35,36].

#### 2.2.3. Increasing the Production of Phenolic Acids by Adding Phenylalanine

Research has proven that addition of phenylalanine to culture medium increases the production of phenolic acids. The total content of phenolic acids present in the methanol extracts from suspension cultures (control and culture supplemented with phenylalanine) are presented in Table 2. As shown in the table, all experimental samples exhibited higher levels of total phenolic acids compared with control samples (7 to 134%). Cultures that were grown for 2 weeks without phenylalanine supplementation (collected after 2 weeks of phenylalanine supplementation) and cultures grown for 3 weeks (collected after 2, 4, and 7 days of phenylalanine supplementation) yielded the best results. However, cultures that were grown for 5 weeks (i.e., 3 weeks without phenylalanine and the following 2 weeks with phenylalanine supplementation) were not found to be suitable for further research due to the drastic decrease in phenolic acid content. This means that cell cultures grown for about 4 weeks possessed the highest biosynthetic potential. Cultures that were maintained with phenylalanine supplementation for nearly 3 weeks showed stronger and better results. On the other hand, phenylalanine supplementation did not affect the biomass growth (Table 3). Maximal content of total phenolic acids (73.76 mg/100 g dry weight (DW) was observed in extracts obtained from cultures that were maintained for 3 weeks and collected after supplementation with the highest concentration of phenylalanine for 4 days. The results obtained for cultures that were supplemented with same phenylalanine concentration and maintained for the same growing period, collected after 2 and 7 days of supplementation (69.81 and 69.57 mg/100 g DW, respectively), were comparable. Phenylalanine, besides tyrosine, is the basic precursor of phenolic metabolites in plants. Among the other metabolites, conversion of this specific aromatic amino acid via shikimate pathway results in the formation of hydroxycinnamic acids [37,38], which, in turn, due to shortening of side chains, results in the formation of hydroxybenzoic acids [39,40]. Which group of metabolites is formed from shikimic acid depends on the species-specific enzyme potential. In the case of *G. biloba* in vitro culture cells, these are benzoic acid derivatives.

The results obtained showed the presence of *p*-hydroxybenzoic, protocatechuic, and gallic acids in all the samples. The major metabolite detected was protocatechuic acid. This metabolite is, like other phenolic acids, a well-known natural antioxidant and is responsible for the scavenging of free radicals, inhibition of the formation of reactive oxygen or nitrogen species, and chelation of metal ions [16,41]. Thus, it exhibits antibacterial and antiviral properties, as well as nephroprotective, hepatoprotective, and neuroprotective properties. It can also be used in the prevention of cardiovascular diseases and cancer [42,43,44]. Gallic acid, another metabolite detected in the extract, possesses additionally strong astringent and antiseptic properties [45]. The results also showed that the highest amounts of protocatechuic acid (ca. 42–47 mg/100 g DW; 75 to 196% higher than the corresponding control samples) were detected in biomass extracts supplemented with 200 mg/150 mL of phenylalanine (Figure 4 and Figure 5). The highest amounts of gallic acids were detected in extracts from biomass supplemented with 200 mg/150 mL of phenylalanine, grown for 2 and 3 weeks. Increases in 2-week cultures ranged from 24 to 209% of the corresponding control, and in 3-week cultures from 41 to 136% (Figure 4 and Figure 5). The effect of phenylalanine concentration on *p*-hydroxybenzoic acid levels was the least noticeable. The highest content of this metabolite, about 9–10 mg/100 g DW, was obtained mainly after administration of 150 and 200 mg/150 mL of phenylalanine (21–84% higher than the corresponding control). Administration of 100 mg of precursor resulted in a 36 to 60% increase in the *p*-hydroxybenzoic acid content (Figure 4 and Figure 5).

### 2.3. Brine Shrimp (Artemia salina) Lethality Bioassay

The brine shrimp, *A. salina*, is an invertebrate utilized for the preliminary assessment of toxicity of bioactive compounds and plant extracts. The speed, cost-effectiveness, and ease of use are the main advantages of using brine shrimp lethality assay. This assay may also be considered an alternative to in vitro cell culture [46] and in vivo assays. Previous studies have shown a good correlation between the results of the oral acute toxicity determination in murine model and this bioassay, suggesting it as a useful tool for predicting acute toxicity of plant extracts [47]. Furthermore, this bioassay has shown good correlation with antitumor activity and can be used as a pre-screening tool for antitumor bioactive compounds [48]. Since there are reports of the cytotoxicity of *Ginkgo biloba* leaf extract [49], *Artemia salina* lethality bioassay was performed to evaluate the potential toxicity of *G. biloba* cell culture extract. The results revealed that the extract was non-toxic against brine shrimps (LC_50_ > 1000 μg/mL).

## 3. Materials and Methods

### 3.1. Chemicals and Solvents

MeOH, chloroform, and glacial acetic acid of analytical grade were purchased from Chempur. Water was purified by a Millipore water purification system. MeOH of HPLC grade was purchased from Merck. FeCl_2_ was obtained from Carlo Erba. Unless indicated otherwise, all chemicals were purchased from Sigma-Aldrich. Caffeic acid, chlorogenic acid, cinnamic acid, elagic acid, gallic acid, gentizic acid, isoferulic acid, neochlorogenic acid, *o*-coumaric acid, protocatechuic acid, rosmarinic acid, salicylic acid, sinapic acid, and syringic acid standards used for the HPLC analysis were purchased from Sigma Aldrich; *p*-coumaric acid, vanillic acid, ferulic acid, and *p*-hydroxybenzoic acid from Fluka; and cryptochlorogenic acid, isochlorogenic acid, and 4-O-feruloylquinic acid from ChromaDex.

### 3.2. In Vitro Cultures

Female leaf explants collected from the Botanical Garden of the Jagiellonian University in Cracow were utilized to establish *G. biloba* callus cultures. Leaf explants (ca. 1-cm long) were surface-sterilized and inoculated onto a solidified MS (Murashige and Skoog) medium [50] containing 2,4-D (2 mg/L) and BA (0.5 mg/L) and maintained at 25 °C under continuous illumination (cool-white fluorescent light, 40 µmol/m^2^ s) to induce calli. The tissue was subcultured every 4 weeks by inoculating ca. 1.0 g fresh weight of the callus onto the fresh MS agar medium containing picloram (4 mg/L) and BA (2 mg/L).

Cell cultures of *G. biloba* were derived from callus cultures (4 g of fresh biomass/25 mL of medium). Agitating cell cultures were grown in a culture room at 25 °C under continuous cool-white fluorescent light and subcultured every 4 weeks onto MS medium containing picloram (4 mg/L) and BA (2 mg/L). For phytochemical investigation, the biomass of cell cultures was collected at the end of the culture period and lyophilized. Cell cultures were carried out in 300 mL Erlenmeyer flasks and shaken at 140 rpm in a rotary shaker (Altel), at 25 °C ± 2 °C, under cool-white fluorescent light, (40 µmol/m^2^ s). Then, 125 mL of MS medium containing picloram (4 mg/L) and BA (2 mg/L) and 10 g of cell fresh weight were added to the Erlenmeyer flasks. After 2 and 3 weeks of cultivation, phenylalanine solution at concentrations of 100, 150, or 200 mg/150 mL medium was added to the cultures. Sterile distilled water was added in parallel to the controls. Samples were collected 2, 4, 7, and 14 days after the addition of phenylalanine. The fresh mass of cell cultures was collected at the end of the culture period and lyophilized. Results were expressed as mean values (*n* = 3) ± standard deviation SD (Excel Microsoft 365).

### 3.3. Content of Secondary Metabolites

#### 3.3.1. Total Phenolic, Flavonoid, and Condensed Tannin Content

The total phenolic content of the methanol extract obtained from *Ginkgo biloba* culture was measured by the Folin-Ciocalteu assay as previously reported [51]. Briefly, 100 μL of solution containing appropriate concentration of each MeOH extract was mixed with 200 μL of Folin-Ciocalteu reagent, 2 mL of distilled water, and 1 mL of 15% sodium carbonate and incubated at room temperature in the dark for 2 h. Then, the absorbance was measured by spectrophotometer at 765 nm. Gallic acid was used as a standard, and the total phenolic content was expressed as mg GAE/g extract (DW) ± SD.

The total flavonoid content of the extracts was measured by the aluminum chloride colorimetric assay [51]. Briefly, 500 μL of each appropriately diluted sample solution was mixed with 1.5 mL of MeOH, 100 μL of 10% aluminum chloride, 100 μL of 1 M potassium acetate, and 2.8 mL of distilled water. The samples were incubated at room temperature in the dark for 30 min, and the absorbance of the reaction mixture was measured at 415 nm. Quercetin was used to make the calibration curve, and the total flavonoid content was expressed as mg QE/g extract (DW) ± SD.

The condensed tannin content of the extracts was determined by the vanillin method as previously described [51]. Briefly, 50 μL of each sample solution was mixed with 1.5 mL of 4% vanillin in MeOH and 750 μL of concentrated hydrochloric acid. After incubation at room temperature in the dark for 20 min, the absorbance of the reaction mixture was measured at 500 nm. (+)-Catechin was used to make the calibration curve, and the condensed tannin content was expressed as mg CE/g extract (DW) ± SD.

The spectrophotometric results were calculated from the average of three independent experiments.

#### 3.3.2. RP-HPLC Analysis

HPLC analysis was used to determine the metabolite content of methanol extracts (sonication, 30 °C, 20 min, three times) obtained from the leaves, callus cultures, and cell cultures. RP-HPLC analysis was carried out as described elsewhere [52] on Merck-Hitachi liquid chromatograph (LaChrom Elite) equipped with a DAD detector L-2455 and Purospher^®^ RP-18e (250 × 4 mm/5 mm) column. Analysis was carried out at 25 °C, with a mobile phase consisting of methanol (A), and methanol:0.5% acetic acid 1:4 (*v*/*v*) (B). The gradient was as follows: 100% B for 0–20 min; 100–80% B for 20–35 min; 80–60% B for 35–55 min; 60–0% B for 55–70 min; 0% B for 70–75 min; 0–100% B for 75–80 min; and 100% B for 80–90 min, at a flow rate of 1 mL/min. Quantification was done by measuring the peak area with reference to a standard curve derived from five concentrations (0.03125–0.5 mg/mL). An exemplary chromatogram of an extract from *Ginkgo biloba* cell cultures, along with UV spectra of the analyzed compounds, as well as chromatograms and UV spectra of standard substances, are included in the Appendix A.

#### 3.3.3. Extraction and Fractionation of Extracts

Freeze-dried biomass of cell cultures (30 g) was exhaustively extracted with methanol at room temperature on a shaker. The extract (4.5 g) was fractionated on a silica gel column (Kieselgel 0.2–0.5 mm, Merck, Art. No. 7733), suspended in hexane, using the following eluents in succession: 1. n-hexane; 2. n-hexane:ethyl acetate (4:1 *v*/*v*); 3. n-hexane:ethyl acetate (1:1 *v*/*v*); 4. ethyl acetate; 5. ethyl acetate:methanol (19:1 *v*/*v*); and 6. methanol. Methanol fractions were further purified according to Spitaler et al. [53]. After dissolving in a mixture of water and methanol (2:1 *v*/*v*), these fractions were shaken in a butanol-separating funnel. The butanol layer was evaporated and then dissolved in methanol and cooled at −10 °C for 24 h. After centrifugation, the supernatant extract was decanted and evaporated. Further purification of the fractions was performed by paper chromatography (Whatman 3, Art. No. 3003 917) using 15% aqueous acetic acid (*v*/*v*) as mobile phase. Finally, the compounds were isolated by preparative thin layer chromatography (PTLC). PTLC of individual fractions was performed using cellulose plates (DC-Alufolien Cellulose, layer thickness 0.1 mm, Merck Art. No. 5552). The following mobile phases were used to develop chromatograms using an ascending technique at room temperature: chloroform:methanol (9:1 *v*/*v*) and chloroform:methanol (17:3 *v*/*v*). The developed chromatograms were viewed under UV lamp light (254 and 365 nm). The strands of compounds were cut from the plates and ground. Then, the compounds were extracted several times using methanol at room temperature. After evaluating by HPLC (chromatograms and UV spectra are included in the Appendix A), fractions containing the same compounds were pooled, dissolved in MeOD, and analyzed by ^1^H NMR method (AMX 500 Bruker ^1^H 600,13 MHz). The compounds (compound 1—2.8 mg; UV max 221, 259, and 294 nm; Rf _chloroform:methanol 17:3_ 0.24; compound 2—1.6 mg, UV max 213, 254 nm; and Rf _chloroform:methanol 17:3_ 0.57) were identified based on ^1^H NMR spectrum (Appendix A) and compared with the spectrum available in the SDBS database (Spectral Database for Organic Compounds) [54].

### 3.4. Antioxidant Activity

#### 3.4.1. Free Radical Scavenging Activity

The DPPH method [55] was used to determine the free radical scavenging activity of the *G. biloba* cell culture extracts. The extract was tested at different concentrations (0.0625–2 mg/mL). An aliquot (0.5 mL) of methanol solution containing different amounts of sample solution was added to 3 mL of daily prepared methanol DPPH solution (0.1 mM). The changes in optical density were measured at 517 nm 20 min after the initial mixing, with a model UV-1601 spectrophotometer (Shimadzu). BHT was used as reference.

The scavenging activity of the sample was measured as the decrease in absorbance compared to the DPPH standard solution. Results were expressed as radical scavenging activity percentage (%) of the DPPH, according to the formula ((A_o_ − A_c_)/A_o_) × 100, where A_o_ is the absorbance of the control and A_c_ is the absorbance in the presence of the sample or standard.

The results obtained from the average of three independent experiments were reported as mean radical scavenging activity percentage (%) ± SD and mean IC_50_ ± SD.

#### 3.4.2. Reducing Power Assay

The reducing power of the *G. biloba* cell culture extract was evaluated by spectrophotometric detection of Fe^3+^–Fe^2+^ transformation method [56]. The extracts were tested at different concentrations (0.0625–2 mg/mL). Different amounts of samples in 1 mL solvent were mixed with 2.5 mL of phosphate buffer (0.2 M, pH 6.6) and 2.5 mL of 1% potassium ferrocyanide [K_3_Fe(CN)_6_]. The resulting mixture was incubated at 50 °C for 20 min and then cooled rapidly, mixed with 2.5 mL of 10% trichloroacetic acid, and centrifuged at 3000 rpm for 10 min. The supernatant obtained (2.5 mL) was mixed with 2.5 mL of distilled water and 0.5 mL of 0.1% fresh ferric chloride (FeCl_3_), and the absorbance was measured at 700 nm after 10 min. An increase in the absorbance of the reaction mixture was considered to indicate an increase in reducing power. An equal volume (1 mL) of water mixed with the solution prepared as described above served as blank. Ascorbic acid and BHT were used as references. The results obtained from the average of three independent experiments were expressed as mean absorbance values ± SD. The reducing power was also expressed as ASE/mL.

#### 3.4.3. Ferrous Ions (Fe^2+^) Chelating Activity

The Fe^2+^ chelating activity of the *G. biloba* cell culture extract was estimated by analyzing the formation of the Fe^2+^-ferrozine complex, according to the method previously reported [57]. The extracts were tested at different concentrations (0.0625–2 mg/mL). Briefly, different concentrations of each sample in 1 mL of solvent were mixed with 0.5 mL of methanol and 0.05 mL of 2 mM FeCl_2_. The reaction was initiated by the adding 0.1 mL of 5 mM ferrozine. Then, the mixture was shaken vigorously and was allowed to stand at room temperature for 10 min. The absorbance of the solution was measured spectrophotometrically at 562 nm. The FeCl_2_ and ferrozine complex molecules served as the control. EDTA was used as reference. The percentage of inhibition of the ferrozine-Fe^2+^ complex formation was calculated by the formula ((A_o—_A_c_)/A_o_) × 100, where A_o_ is the absorbance of the control and A_c_ is the absorbance in the presence of the sample or standard. The results obtained from the average of three independent experiments were expressed as mean inhibition of the ferrozine-Fe^2+^ complex formation (%) ± SD and IC_50_ ± SD.

### 3.5. Brine Shrimp (A. salina) Lethality Bioassay

The potential toxicity of *G. biloba* cell culture extract was investigated by determining the median lethal concentration (LC_50_) according to the method previously reported [58]. The extract, opportunely dissolved and then diluted in artificial seawater, was tested at final concentrations of 10, 100, 500, and 1000 µg/mL. Ten brine shrimp larvae were transferred to each sample vial, and artificial seawater was added to obtain a final volume of 5 mL. After 24 h of incubation (25–28 °C), the number of larvae that survived was counted. The assay was carried out in triplicate, and LC_50_ values were determined by the Litchfield and Wilcoxon method. Extracts were considered nontoxic if the LC_50_ was higher than 1000 µg/mL.

### 3.6. Statistical Analyses

All statistical analyses were conducted using the STATISTICA 13.3 software program (TIBCO Software Co., Palo Alto, CA, USA). The level of significance was set at *p* < 0.05. The differences in results across the groups were analyzed two-way variance analysis, followed by a Bonferroni post hoc test. The results were expressed as means ± SD of the mean.

## 4. Conclusions

Currently, the use of plant in vitro cultures for cosmetological purposes is becoming increasingly common. The cosmetic ingredient database (CosIng) contains a lot of information on these substances and their functions. *Ginkgo biloba* callus culture extract is recommended as a skin conditioning while, and *G. biloba* meristem cell culture extract as an antimicrobial, antioxidant, and skin conditioning [59].

These studies confirmed the antioxidant activity of *G. biloba* cell culture extract, indicated the phenolic acids as the main compounds, and showed that all the phenolic acids detected in the biomass of *G. biloba* cell culture are benzoic acid derivatives, which means that the entire biosynthetic pathway of phenolic acids is redirected to the synthesis of this group of phenolics. The results also revealed that phenylalanine supplementation enhanced this process, which will allow obtaining more valuable materials from in vitro cultures. The best conditions for obtaining the highest content of phenolic acids were supplementation with 200 mg of phenylalanine after 3 weeks of in vitro culture and collecting after 4 days. Phenylalanine is the basic precursor to the metabolic pathway of phenolic metabolites in plants. In *Ginkgo biloba* cell cultures, it is mainly converted to phenolic acids, which shows that the next stages in the metabolic pathway may be impaired or blocked.

## Figures and Tables

**Figure 1 molecules-26-04965-f001:**
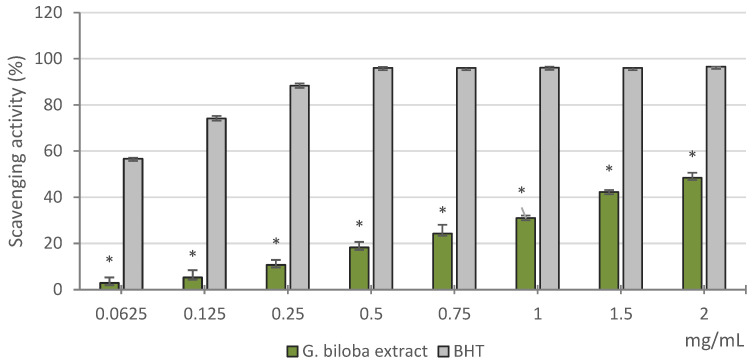
Free radical scavenging activity (DPPH assay) of the methanol extract obtained from *G. biloba* cell culture. Values are expressed as the mean ± SD (*n* = 3). * *p* < 0.05 vs. BHT.

**Figure 2 molecules-26-04965-f002:**
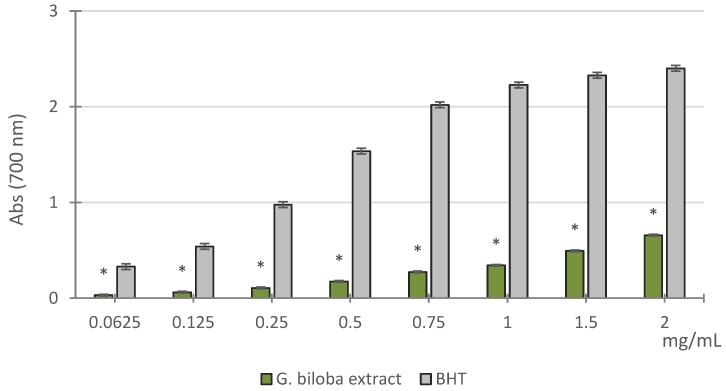
Reducing power of the methanol extract obtained from *G. biloba* cell culture. Values are expressed as the mean ± SD (*n* = 3). * *p* < 0.05 vs. BHT.

**Figure 3 molecules-26-04965-f003:**
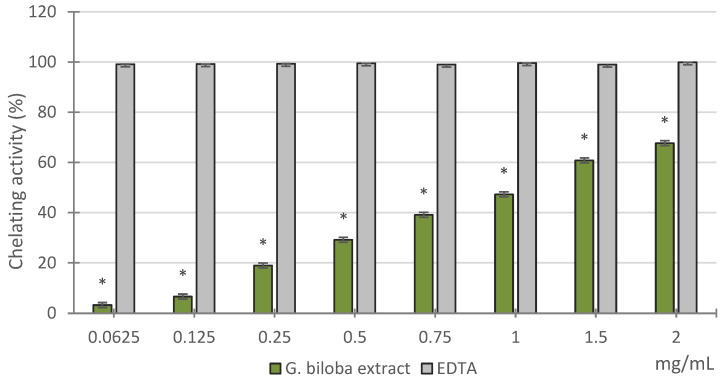
Ferrous ions (Fe^2+^ chelating activity of the methanol extract obtained from *G. biloba* cell culture). Values are expressed as the mean ± SD (*n* = 3). * *p* < 0.05 vs. BHT.

**Figure 4 molecules-26-04965-f004:**
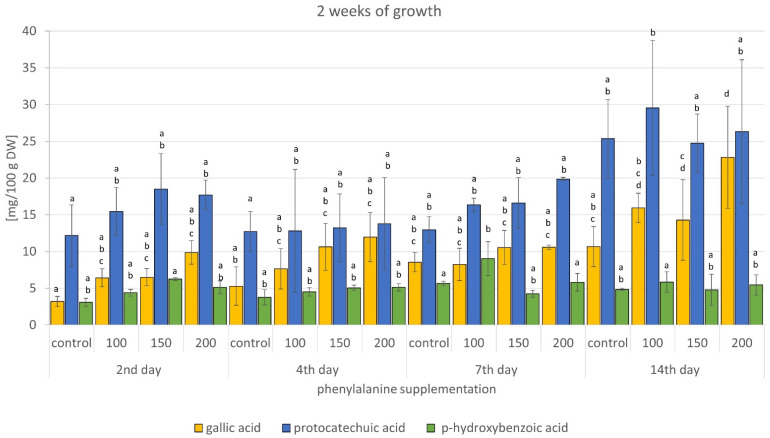
Content of phenolic acids in *G. biloba* cell culture extracts supplemented with phenylalanine—2 weeks of growth. Different letters above bars indicate significant differences for each phenolic acid (*p* < 0.05).

**Figure 5 molecules-26-04965-f005:**
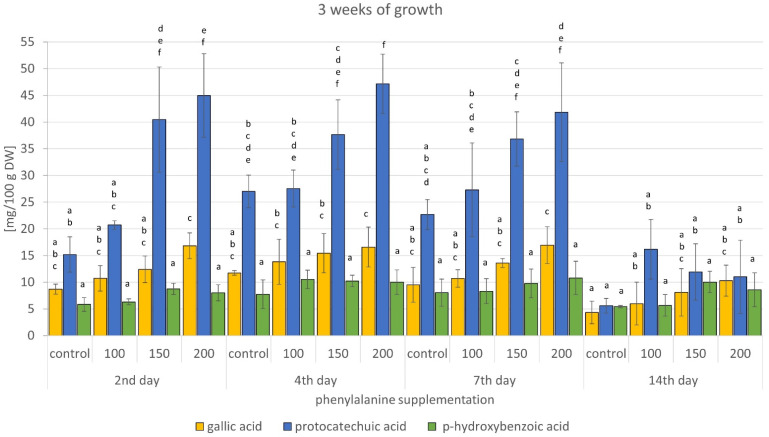
Content of phenolic acids in *G. biloba* cell culture extracts supplemented with phenylalanine—3 weeks of growth. Different letters above bars indicate significant differences for each phenolic acid (*p* < 0.05).

**Table 1 molecules-26-04965-t001:** Accumulation of phenolic acids in the methanol extracts obtained from *G. biloba* in vitro cultures and leaves (mg/g DW); means of three measurements ± SD.

Methanol Extract of	*p*-Hydroxybenzoic Acid	Protocatechuic Acid	Gallic Acid	Vanillic Acid
cell	0.043 ± 0.0066	0.2118 ± 0.0025	0.0078 ± 0.0002	0.002 ± 0.00006
callus	0.0417 ± 0.002	0.2324 ± 0.0035	0.0123 ± 0.0014	0.0194 ± 0.0002
leaves	0.1263 ± 0.0084	0.235 ± 0.046	0.0212 ± 0.0018	0.0445 ± 0.0055

**Table 2 molecules-26-04965-t002:** Effect of phenylalanine supplementation on phenolic acid production (mg/100 g DW); means of three measurements ± SD. Different letters indicate significant differences (*p* < 0.05).

Growing Period before Supplementation of Phenylalanine	Collecting Time after Supplementation of Phenylalanine	Phenylalanine (mg/150 mL)	Total Phenolic Acids (mg/100 g DW)
2 weeks	2nd day	0 (control)	18.461 ± 4.963 ^a,b^
		100	26.253 ± 2.870 ^a,b,c,d,e^
		150	31.270 ± 4.248 ^a,b,c,d,e,f,g^
		200	32.705 ± 0.819 ^a,b,c,d,e,f,g^
	4th day	0 (control)	21.787 ± 4.836 ^a,b,c^
		100	24.975 ± 10.851 ^a,b,c,d^
		150	28.894 ± 3.440 ^a,b,c,d,e^
		200	30.902 ± 3.934 ^a,b,c,d,e,f^
	7th day	0 (control)	27.176 ± 3.386 ^a,b,c,d,e^
		100	33.618 ± 4.119 ^a,b,c,d,e,f,g^
		150	31.399 ± 1.404 ^a,b,c,d,e,f,g^
		200	36.274 ± 1.092 ^a,b,c,d,e,f,g,h^
	14th day	0 (control)	40.871 ± 8.091 ^c,d,e,f,g,h,i,j^
		100	51.325 ± 8.210 ^f,g,h,i,j,k,l^
		150	43.832 ± 7.320 ^d,e,f,g,h,i,j,k^
		200	54.601 ± 5.846 ^h,i,j,k,l,m^
3 weeks	2nd day	0 (control)	29.736 ± 4.574 ^a,b,c,d,e^
		100	37.777 ± 1.649 ^b,c,d,e,f,g,h^
		150	61.629 ± 10.004 ^j,k,l,m^
		200	69.805 ± 7.951 ^l,m^
	4th day	0 (control)	46.482 ± 4.692 ^e,f,g,h,i,j,k^
		100	51.926 ± 9.194 ^g,h,i,j,k,l^
		150	63.313 ± 10.154 ^k,l,m^
		200	73.761 ± 5.570 ^m^
	7th day	0 (control)	40.246 ± 1.869 ^c,d,e,f,g,h,i^
		100	46.319 ± 8.935 ^e,f,g,h,i,j,k^
		150	60.175 ± 2.564 ^i,j,k,l,m^
		200	69.566 ± 8.961 ^l,m^
	14th day	0 (control)	15.390 ± 3.646 ^a^
		100	27.811 ± 1.896 ^a,b,c,d,e^
		150	30.041 ± 7.386 ^a,b,c,d,e^
		200	29.925 ± 8.983 ^a,b,c,d,e^

**Table 3 molecules-26-04965-t003:** Effect of phenylalanine supplementation on biomass growth (g); means of three measurements ± SD. Different letters indicate significant differences (*p* < 0.05).

Growing Period before Supplementation of Phenylalanine	Collecting Time after Supplementation of Phenylalanine	Phenylalanine (mg/150 mL)	Dry Weight (g)
2 weeks	2nd day	0 (control)	0.539 ± 0.039 ^a^
		100	0.632 ± 0.061 ^a,b^
		150	0.678 ± 0.080 ^a,b^
		200	0.687 ± 0.061 ^a,b^
	4th day	0 (control)	0.818 ±0.069 ^b,c,d,e^
		100	0.774 ± 0.062 ^a,b,c,d^
		150	0.960 ± 0.135 ^c,d,e,f,g^
		200	0.759 ± 0.014 ^a,b,c^
	7th day	0 (control)	0.876 ± 0.065 ^b,c,d,e,f^
		100	1.082 ± 0.168 ^e,f,g^
		150	1.034 ± 0.010 ^d,e,f,g^
		200	1.197 ± 0.069 ^g,h,i^
	14th day	0 (control)	1.472 ± 0.006 ^j^
		100	1.373 ± 0.044 ^h,i,j^
		150	1.424 ± 0.066 ^i,j^
		200	1.410 ± 0.044 ^i,j^
3 weeks	2nd day	0 (control)	1.065 ± 0.072 ^e,f,g^
		100	1.063 ± 0.074 ^e,f,g^
		150	1.099 ± 0.107 ^f,g^
		200	1.132 ± 0.074 ^f,g,h^
	4th day	0 (control)	1.436 ± 0.107 i,j
		100	1.380 ± 0.022 ^h,i,j^
		150	1.520 ± 0.030 ^j^
		200	1.470 ± 0.125 ^j^
	7th day	0 (control)	1.490 ± 0.049 ^j^
		100	1.467 ± 0.034 ^i,j^
		150	1.450 ± 0.109 ^i,j^
		200	1.443 ± 0.046 ^i,j^
	14th day	0 (control)	1.446 ± 0.068 ^i,j^
		100	1.400 ± 0.076 ^h,i,j^
		150	1.378 ± 0.094 ^h,i,j^
		200	1.478 ± 0.156 ^j^

## Data Availability

Not applicable.

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
