# Peer review of "Phenylalanine Increases the Production of Antioxidant Phenolic Acids in Ginkgo biloba Cell Cultures"

_molecules, 2021, doi:10.3390/molecules26164965_

Round 1

Reviewer 1 Report

Dear authors,

Your submitted paper contains an intresting approach in order to investigate partial metabolic benzoic acid changes in G. Biloba cultures, which is supported by sufficient experimental methodology.

However, your results are not presented very comprehensive in terms of protocatechuic/p-hydroxy benzoic acid content etc simple numerical differences (%) before and after Phe addition, in relation too RP_HPLC data. This lack is more obvious  in conclusion section, e.g., DPPH and chelating activities, culture time which yielded best results,  type of derivatives with the highest culture production.

Furthermore, some discussion is also needed for the other important antioxidant compounds e.g flavonoids/polyphenolic content  difference

As for the Artemia salina test, see also ref: International Journal of Pharmacognosy (Pharmaceutical Biology), 34(1), 19-27, 1996  

Author Response

Dear Reviewer,

We are greatly obliged for having received the Reviewers’ valuable opinion and helpful suggestion on our manuscript. All changes in the manuscript are marked in yellow. The replies to the specific comments are listed below.

‘Your submitted paper contains an intresting approach in order to investigate partial metabolic benzoic acid changes in G. Biloba cultures, which is supported by sufficient experimental methodology.’

Thank you for your kind and supportive words about our paper.

However, your results are not presented very comprehensive in terms of protocatechuic/p-hydroxy benzoic acid content etc simple numerical differences (%) before and after Phe addition, in relation too RP_HPLC data. This lack is more obvious  in conclusion section, e.g., DPPH and chelating activities, culture time which yielded best results,  type of derivatives with the highest culture production.’

We would like to explain that all data (except biomass growth) presented in chapter 2.2.3. was obtained by HPLC analysis. The text was supplemented with information about the increase in content of phenolic acids. Numerical differences in % was given for the most important results. Data with the best results were added in conclusion.

‘Furthermore, some discussion is also needed for the other important antioxidant compounds e.g flavonoids/polyphenolic content  difference’

We have improved the discussion of the results of the three antioxidant tests in relation to the contents of polyphenols determined.

‘As for the Artemia salina test, see also ref: International Journal of Pharmacognosy (Pharmaceutical Biology), 34(1), 19-27, 1996’ 

We have added a sentence referred to the article suggested by the reviewer in the section “2.3. Brine shrimp (Artemia salina) lethality bioassay”.

Please, accept my best regards,

Yours sincerely,

Agnieszka Szewczyk

Reviewer 2 Report

In this study, the results showed that  supplementation of phenylalanine to the culture medium as a metabolites precursor could be a strategy for obtained valuable secondary metabolites using in vitro cultures. For obtained an manuscript with more quality, I would like to suggest that paragraph be included in introduction for describe the main characteristics of secondary metabolites production using in vitro cultures in diverse plant species, in special for cosmetology purposes.

Line 59- 69. delete paragraph or change for introduction

line 122-130 also delete paragraph.

in all document change in vitro for cursives format

Author Response

Dear Reviewer,

We are greatly obliged for having received the Reviewers’ valuable opinion and helpful suggestion on our manuscript. All changes in the manuscript are marked in yellow. The replies to the specific comments are listed below.

‘In this study, the results showed that  supplementation of phenylalanine to the culture medium as a metabolites precursor could be a strategy for obtained valuable secondary metabolites using in vitro cultures. For obtained an manuscript with more quality, I would like to suggest that paragraph be included in introduction for describe the main characteristics of secondary metabolites production using in vitro cultures in diverse plant species, in special for cosmetology purposes.’

We are grateful for this valuable suggestion. We would like to explain that Introduction section was supplemented with information regarding cosmetology purposes of in vitro cultures.

‘Line 59- 69. delete paragraph or change for introduction’

We deleted the paragraph and changed the introduction.

‘line 122-130 also delete paragraph.’

We deleted from line 122 to 126 and we changed the text in the paragraph.

‘in all document change in vitro for cursives format’

We changed “in vitro” in cursive format in all the document.

Please, accept my best regards,

Yours sincerely,

Agnieszka Szewczyk

Reviewer 3 Report

The article is interesting, extracts of Ginkgo and biloba have been extensively studied in the literature, no plagiarism or any serious problem is detected, but if the direction of the experiments disconcerts me, especially the part of the isolation and characterization of substances, It seems to me that these were not well designed and placing a spectrum of ready mixtures from the extracts cannot be considered as isolation, although they are known substances, it is also true that to give this type of evidence the substances must be isolated and properly purified . , and given the evidence, it seems that is not the case. Finally, I enclose the revised file with several observations, which can be analyzed and if possible considered, a cordial greeting.

Author Response

Dear Reviewer,

We are greatly obliged for having received the Reviewers’ valuable opinion and helpful suggestion on our manuscript. All changes in the manuscript are marked in yellow. The replies to the specific comments are listed below.

‘The article is interesting, extracts of Ginkgo and biloba have been extensively studied in the literature, no plagiarism or any serious problem is detected, but if the direction of the experiments disconcerts me, especially the part of the isolation and characterization of substances, It seems to me that these were not well designed and placing a spectrum of ready mixtures from the extracts cannot be considered as isolation, although they are known substances, it is also true that to give this type of evidence the substances must be isolated and properly purified , and given the evidence, it seems that is not the case.’

We would like to explain that the main aim of the study was to investigate how the addition of phenylalanine would affect the production of phenolic acids. Analyzes were performed by HPLC with DAD detection. We agree with the note that phenolic acids are well known compounds. In our opinion, UV spectra would confirm the identity of these compounds. But often reviewers ask for identity confirmation by spectral methods. Isolation of compounds from such a difficult material as in vitro culture material raises many problems. For economic reasons, we cannot obtain a large amount of material for research. In the case of Ginkgo biloba cell cultures, we managed to obtain 30 g. At each stage of isolation, a lot of compounds are lost, so we could only isolate the main components. Each isolation step was controlled by HPLC, which showed a fairly high purity of the fractions obtained. In Supplementary file, we include chromatograms of the isolated compounds.

‘Finally, I enclose the revised file with several observations, which can be analyzed and if possible considered, a cordial greeting.’

We followed the suggestions reported in the pdf file:

Page 2:

‘This is introduction, not results or introduction, please change it.’

We are grateful for this valuable suggestion. We deleted this paragraph in this section and shifted it in the section Introduction.

Pages 3-4:

‘This concept is very important and I think it should be broadened and highlighted.’

We would like explain that we wrote a more extensive comment to highlight this part.

The discussion of the results is poor please improve’

We would like explain that we improved the discussion of the results of the three antioxidant tests in relation to the content of polyphenols determined.

‘The image quality is poor, please improve the presentation, I suggest including statistical analysis of significance.

We replaced the figures 1, 2, and 3 with better quality images; moreover, we added in all the graphs the statistical analysis.

Page 5:

‘Incomprehensible paragraph please rephrase.’

We modified the paragraph to make it clearer.

‘The isolation and characterization of the metabolites is poor, although the proton spectra are shown in the supplementary material the identification of 4 metabolites are mentioned in table 1, the spectra presented are dirty, no purification, carbon 13 spectra are missing for confirmation of carboxylic acid groups.’

We would like explain that 1HNMR analyzes were performed as additional confirmation of the identity of the compounds analyzed by HPLC. In the Supplementary file, we added chromatograms and UV spectra of the isolated compounds. Due to the small amount of material it was possible to isolate only the main compounds, the remaining compounds are present in very small amounts. The content of vanillic acid in cell cultures is trace, so we did not investigate the content of this compound in the analyzes of the main experiment (phenylalanine addition).

Page 6:

‘Emphasis is made on the increase in the concentration of phenolic acids but it is not explained why, I understand that it is due to the increase in exposure to phenylalanine but a satisfactory explanation is not given of why this phenomenon occurs.’

We would like explain that we have added an explanation about the metabolic pathway of phenolic acids and the role of phenylalanine.

Page 8:

‘The image quality is poor, please improve the presentation, describes the detection of three metabolites, but in the supplementary material there is only evidence of two, how were they detected, how did it distinguish one from the other?’

We replaced the figures 4 and 5 with better quality images. In the supplementary materials, we added chromatograms of the extract and reference substances along with UV spectra.

‘toxicity bioassay’

We disagree with this suggestion; indeed, we have a great deal of experience with this test, which we have been conducting for many years and, in our opinion, it is appropriate to indicate it as “Artemia salina lethality bioassay”.

Page 10:

‘please include histograms as evidence of HPLC performance.’

In the supplementary materials, we added chromatograms of the extract and reference substances along with UV spectra.

‘Please indicate the Rf of the substances, and the yields of the substances’

We would like to explain that the text was supplemented with this information.

Page 12:

‘The conclusions should be improved, please, a brief and concise explanation of the observed phenomenon’

We would like to explain that Conclusion section has been changed.

We would like to express our appreciation for Reviewers’s time and efford.

Please, accept my best regards,

Yours sincerely,

Agnieszka Szewczyk
